# Affiliation in times of pandemics: Determinants and consequences

**Guillaume Dezecache**[1,2]* *, **Johann Chevalère**[1]*, **Natalia Martinelli**[1], **Sandrine Gil**[3], **Clément Belletier**[1], **Sylvie Droit-Volet**[1], **Pascal Huguet**[1]

1 Université Clermont Auvergne, LAPSCO, CNRS, Clermont-Ferrand, France, 2 UMI SOURCE, Université Paris-Saclay, UVSQ, IRD, Guyancourt, France, 3 Université de Poitiers, CeRCA, CNRS, Poitiers, France

☉ These authors contributed equally to this work.
* guillaume.dezecache@ird.fr

## Abstract

Affiliation is a basic human need, especially during difficult times. To what extent did the need to affiliate limit our capacity to abide by health guidelines, in particular regarding social distancing, during the COVID-19 pandemic? We investigated this issue using questionnaire data from two samples of the French population collected during the first French lockdown (April-May 2020). We found that in men, higher social comparison orientation (sensitivity to the needs of others and inclination to help) and higher perceived threat increased the frequency of reported affiliative activities. At the same time, men's reported affiliative activities were associated with a lower reported intention to abide by lockdown and protective measures and lower levels of reported compliance. This pattern was not found in women. The women in our samples, as has been observed elsewhere, were largely compliant, potentially precluding any effects of affiliative needs. Basic though they may seem, affiliative needs and reported affiliative activities may have played a significant role in the implementation of sanitary guidelines during the COVID-19 pandemic.

## Introduction

Climate change is increasing the risk of the zoonotic transmission of pathogens from nonhuman animals to humans [1]. Pandemic episodes are therefore expected to occur more regularly. The recent COVID-19 pandemic reminded us of the devastating consequences of pandemic episodes and the challenges they pose to societies [2–6]. When confronted with rapid viral mutations, people should comply with public policy rules or guidelines, such as physical distancing, face masking, hand hygiene and the physical isolation of infected individuals. One central question that has persisted even after the discovery and the global distribution of vaccines is how to ensure compliance. This remains a key issue today, for new viruses do and will continue to emerge [1].

The psychological and behavioral sciences have been at the forefront of policy recommendations [7,8]. Empirical research investigating compliance with public policy rules around COVID-19 has confirmed that social identities and norms are vital factors determining successful implementation (see [9] for a review). One major finding by Tunçgenç and colleagues

**Data Availability Statement:** The data and analysis scripts can be accessed at: https://osf.io/p3rtx/?view_only=f7ecdb41adc949e19d921498e4169e76.

**Funding:** Note that our work was supported by the Agence Nationale de la Recherche (ANR, France)

via the ANR-Flash COVID-19 scheme. The funders had no role in study design, data collection and analysis, decision to publish, or preparation of the manuscript.

**Competing interests:** The authors have declared that no competing interests exist.

[10], based on a large sample of around 6675 participants from 115 countries, is that participants reported adhering more closely to social distancing rules when their close circle (e.g., close friends) also did so. This social norm (what close others do) had a stronger effect than the participants' own approval of the rules or their feelings of vulnerability to the disease. This finding is consistent with decades of research on 'social comparison' processes and their consequences [11–13]. Situations associated with uncertainty (and sometimes marked by anxiety) about what to think and what to do make individuals more sensitive to what close others (friends, family members, loved ones, etc.) and/or those perceived as similar to themselves (same sex, age, etc.) believe and do, and also make them more likely to engage in social comparison.

Although social comparison has traditionally been associated with a motivation to differentiate oneself from others in a competitive way, social comparison corresponds to the process of relying on what others think, feel, and behave in order to form correct assessment and invest in appropriate behavior [14,15]. There are, in fact, two broad, independent dimensions underlying social comparison processes: a 'vertical' (better/worse than others) dimension of status, dominance, or agency [12,16–18]; and a 'horizontal' dimension of solidarity, friendliness, or communion (for a review, see [19]). Originally indeed, 'social comparison' is a process whereby individuals seek to obtain correct evaluations of their opinions and abilities. In the absence of objective standards, they may look at others and compare themselves [14,15]. For instance, when uncertain about what to do in a novel situation or stressful situation, individuals rely on social comparison to evaluate the meaning of the situation, and best assess how they should behave. This is a case of horizontal social comparison. This happens between persons who tend to experience uncertainty and close or similar others favor an increase in contact and affiliative behavior. This was highlighted by Schachter as early as 1959 [20]. Schachter [20] showed that contact and affiliation with others reduce uncertainty and related anxiety. Affiliative and contact-seeking behaviors are also very common among populations in emergency situations [21–25] or people exposed to a threatening stimulus [22]. Such affiliative behaviors could be tightly linked to social comparison processes, whereby being and comparing oneself to others enable more certainty about the situation and optimal behavioral conduct.

While social norms and horizontal social comparison processes are powerful influences that need to be considered when formulating recommendations, they may also undermine the efficient implementation of these recommendations. This was suggested by Dezecache et al. [26], who explained that the affiliative needs we have acquired as a social species to cope with uncertainties and threats may also facilitate the spread of viruses, as physical proximity and contact increase disease transmission. However, whether this link (greater feeling of threat associated with greater affiliative needs, themselves associated with reduced compliance with guidelines) actually holds in practice remains to be determined. Although the tendency to engage in social comparison is thought to be a universal human characteristic—a 'phylogenetically very old' and 'biologically very powerful' tendency [27], there is evidence that its strength varies between individuals [28]. People's need to engage in horizontal social comparison may therefore be a determining factor in their willingness to break sanitary rules to meet their affiliative needs.

Conducted in the context of the COVID-19 pandemic, the present research examined the relationships between individuals' level of horizontal social comparison, their reported affiliative behaviors, and their reported compliance with pandemic guidelines. We expected that higher social comparison orientation (i.e., a greater need to rely on what others think and how they behave) would be associated with a greater need to communicate with others, as a strategy to reduce distress We also expected this association to be exacerbated in threatening circumstances (i.e., fear of the pandemic and infection), thereby increasing affiliative needs and

reducing compliance with guidelines, in particular those restricting social contact outside the household. This has likely played a role given the role friendship and affiliative ties more generally have in reducing distress and feelings of loneliness. The protective role of affiliation with friends in mitigating feelings of loneliness during the COVID-19 pandemic has been highlighted in previous work, particularly in young people [29]. The psychologically protective but physically endangering role 'friendship' and affiliative ties might have played during the COVID-19 pandemic has also been highlighted by De Vries & Lee [30] who experimentally showed that the safety felt from psychological closeness with friends could induce lower perception of infection risks, which could in turn increase risk-taking. Here, we examined Dezecache et al.'s [26] hypothesis while taking account of other potential determinants of compliance. Specifically, we examined gender as a major source of variance in reports of respect for sanitary guidelines. This approach follows on from the work of Galasso and colleagues [31], who questioned a total of 21,649 respondents from 8 countries and found that women were more likely than men to perceive the COVID-19 pandemic as a serious health problem. Women agreed with restrictive public policy measures (social distancing rules) and reported complying with them. These differences could not be accounted for by sociodemographic and employment characteristics or by psychological and behavioral factors. They were only partially mitigated in individuals who cohabited or who were directly exposed to the virus, thereby indicating how powerful gender differences can be when it comes to beliefs, attitudes and behaviors regarding the pandemic and related public policy. It should be noted that the term 'gender' was likely used interchangeably with 'sex roles' in the current manuscript. This choice was made after consideration of the country of study (France) where this conceptual distinction was not particularly common or widespread at the time of the study. Additionally, using the word 'sex' rather than gender could have been interpreted as a claim that we could uncover a difference between 'males' and 'females' (which we had no means to do with the current technique [questionnaires]), which is not the product of gender socialization itself but rather a product of biological differences (and the expression thereof).

Besides gender, we also took account of whether the perception of danger, in the form of one's own perceived vulnerability to the disease [32], and the perception of illness [33] increased threat-related emotions of anxiety and fear [34]. Further justification for taking account of threat-related emotions in the pandemic context comes from documented evidence that women reported higher levels of fear and anxiety than men [35–38] (but see [39] for a discussion of gender differences in anxiety reports in the context of the COVID-19 pandemic), even though the disease was more severe in men [40]. Furthermore, threat-related emotions are a relevant variable in relation to compliance with COVID-19 public policy rules. Ayers and colleagues [41] found reports of greater loneliness and isolation in people anxious about contracting COVID-19, and in particular among young people and those reporting higher subjective SES. Greater feelings of stress and isolation were also associated with higher risk-taking in the social domain.

We applied Structural Equation Modeling (SEM) to two relatively large datasets taken from a larger online study (which also included items on time perception) conducted among the French general population during the lockdown of March 2020—May 2020 [42]. We measured (among other items) compliance intentions (regarding physical distancing), self-reported compliance behaviors, levels of fear and anxiety, time spent in online communication with close others (i.e., affiliative behaviors, reflecting affiliative needs), as well as horizontal social comparison [28]. While most participants in the first sample were women (76.3%), the second was balanced (49.3% men), enabling us to evaluate so-called 'gender' differences in the relationships between horizontal social comparison, reported feeling of threat, affiliative activities and reported respect for sanitary guidelines.

## Methods

### Ethics

All the participants gave informed written consent using the questionnaire. The questionnaire was approved by the Research Ethics Committee of the Université Clermont Auvergne (IRB00011540-2020-31).

### Pre-registration note

Because of the urgency of the situation (the first period of lockdown in France ran from March 2020 to May 2020), the current study was not formally pre-registered. The first sample was recruited using social media. After recognizing the clear gender imbalance in Sample 1, we asked a market research specialist to recruit a second sample (Sample 2). We decided to present the results of both samples (and compare them) to improve transparency and because they make it possible to evaluate replicability. Sample 2 provided us with a more balanced dataset and allowed us to limit 'self-selection' as a sampling bias.

### Participants

Sample 1 initially included 1,299 participants recruited through snowball sampling within our social networks. The study was also published on the University's web site and contained a direct link to the questionnaire. Anyone who saw the link was therefore able to participate. We did not verify the responses regarding how the participants had accessed the link to the questionnaire. Two hundred and fifteen respondents (16.6%) were excluded because they did not fully complete the questionnaire. Due to the considerable gender imbalance in the remaining sample (only 23.7% men out of 1084 participants) and because the number of men in this sample ($N = 257$) was clearly below the minimum sample size ($N = 450$) required to test the structural relations described below, the first analysis was conducted on women only (827 participants; $M_{age} = 40.93$ [17, 92], $SD = 15.94$), hereafter referred to as "Sample 1W". The participants in this sample had an average period of education (number of years of education since first enrolment in primary school in France) of 14.95 years ($SD = 2.87$), ranging from a minimum of 7 years of education, corresponding to a primary school certificate, to a maximum of 20 years of education, corresponding to a doctorate or equivalent degree, with 93% of the participants having obtained at least the high school diploma. 0.07% of them were agricultural workers, 1% tradespeople, shopkeepers or company managers, 15.5% occupied executive or higher intellectual professions, 20.3% intermediate occupations. 11.7% were employees, 0.73% manual workers. 14.7% were retired, and the remaining 36% were without professional activity (including students). Participants provided their post code but no other geographical information. In line with national ethical guidelines and legislation, no information about race and ethnicity could be requested.

To ensure a better gender balance, the initial second sample—1,278 participants—was recruited via a market research company (EasyPanel). Two hundred and forty responders (18.8%) were excluded because they failed to fully complete the questionnaire. The analysis was conducted on 1038 participants ($M_{age} = 45.50$ [18, 75], $SD = 14.94$), with a well-balanced proportion of women (50.7%) and men (49.3%). The participants in this second sample had an average period of education of 13.08 years ($SD = 2.89$), with 76.9% of them having obtained at least the high school diploma. 0.3% of them were agricultural workers, 2.5% tradespeople, shopkeepers or company managers, 11.4% occupied executive or higher intellectual professions, 19.5% intermediate occupations. 25.3% were employees, 9.3% manual workers. 18.7% were retired, and the remaining 13% were without professional activity (again including

students). Participants also provided their post code but no other geographical information. Again, French ethical guidelines and legislation meant that no information about race and ethnicity could be requested.

## Material and procedure

The questionnaire was administered online using LimeSurvey and the data was hosted on a local server. The questionnaire took about 40 minutes to complete and was administered between April-1 2020 and April-29 2020. This coincided with the French lockdown of March-17 2020 to May-11 2020. The questions and scales used in this study can be found in Table 1.

We used the Social Comparison Orientation (SCO) scale [28] to measure horizontal social comparison. This scale measures the inclination and willingness to compare one's accomplishments, situations, or experiences with those of others (the scale thus includes two sub-dimensions, one relating to 'ability' and the other to 'opinion'). The SCO scale includes items such as 'I always pay a lot of attention to how I do things compared with how others do things.' (item 2, 'ability' subscale), and 'I often try to find out what others think who face similar problems as I face' (item 9, 'opinion' subscale). It has been adapted and successfully used in countries as diverse as Hungary, Poland, Turkey [43], the USA, Netherlands, Spain [28,44], and France [45–47], among many others. The SCO scale correlates positively with scales measuring interpersonal orientation (including a strong empathy for others; an interest in mutual self-disclosure) and communal orientation (inclination to care for and help others) [48]. This pattern of correlations (the higher the SCO score, the higher the interdependence and communal values) may seem counter-intuitive since social comparison has traditionally been associated with a motivation to differentiate oneself in a competitive way from others. However, there is now ample evidence that two broad, independent dimensions underlie social comparison processes: a 'vertical' (better/worse than others) dimension of status, dominance, or agency [12,16–18], and a 'horizontal' dimension of solidarity, friendliness, or communion (for a review, see [19]). The SCO scale clearly taps into the horizontal rather than vertical dimension of social comparison.

## Statistical analyses

All statistical analyses were conducted using the R software [49] version R-4.0.2. The SEM framework was applied using the Lavaan package [50] version 0.6–3 to model the structural relationships among a series of psychological constructs which we think determine women's and men's reported compliance in terms of a) staying at home (lockdown) and b) taking protective measures. We were able to examine two types of elements of the structural model. On the one hand, standardized latent means corresponded to estimated values for the unobserved psychological constructs of interest (e.g., affiliative tendencies and compliance with lockdown orders). The latent means were calculated based on the shared variance among a series of observed variables, each thought to reflect the unobserved construct to some extent, thus minimizing the measurement error for the construct [51]. On the other, path coefficients corresponded to the magnitude of the actual relations between the latent variables. Specifically, we examined a model comprising the following latent variables: SCO, Perceived Vulnerability, Perceived Illness, Threat-Related Emotions, Affiliation, Compliance Intentions, and reported Compliance Behavior.

The items from which the latent variables were derived are shown in Figs 1 to 6 and described in detail in Table 1. The baseline levels for the (pre-lockdown) 'fear' and 'anxiety' items were entered as covariates of Threat-Related Emotions (the latent variable, therefore, indicates a change in threat-related emotions from before to during lockdown). It should be

**Table 1. Items used in the structural models and factor loadings.**

| Latent variable | Description | Item | Factor loading Sample 1W | Factor loading Sample 2 | Factor loading Sample 2M | Factor loading Sample 2W |
|---|---|---|---|---|---|---|
| SCO ABI[1] | SCO ability item1: *'I often compare how my loved ones (boy or girlfriend, family members, etc.) are doing with how others are doing.'* from strongly disagree [1] to strongly agree [6] | 1 | .75 | .75 | .75 | |
| SCO ABI[1] | SCO ability item 2: *'I always pay a lot of attention to how I do things compared with how others do thing'* from strongly disagree [1] to strongly agree [6] | 2 | .86 | .80 | .80 | .80 |
| SCO ABI[1] | SCO ability item 3: *'If I want to find out how well I have done something, I compare what I have done with how others have done'* from strongly disagree [1] to strongly agree [6] | 3 | .82 | .84 | .82 | .85 |
| SCO ABI[1] | SCO ability item 4: *'I often compare how I am doing socially (e.g., social skills, popularity) with other people'* from strongly disagree [1] to strongly agree [6] | 4 | .77 | .83 | .84 | .82 |
| SCO ABI[1] | SCO ability item 6: *'I often compare myself with others with respect to what I have accomplished in life'* from strongly disagree [1] to strongly agree [6] | 6 | .73 | .79 | .79 | .78 |
| SCO OPI[1] | SCO opinion item 8: *'I often try to find out what others think who face similar problems as I face'* from strongly disagree [1] to strongly agree [6] | 8 | .76 | .83 | .85 | .82 |
| SCO OPI[1] | SCO opinion item 9: *'I always like to know what others in a similar situation would do'* from strongly disagree [1] to strongly agree [6] | 9 | .88 | .89 | .91 | .87 |
| SCO OPI[1] | SCO opinion item 10: *'If I want to learn more about something, I try to find out what others think about it'* 0—This question does not apply to me, 1 • Very unlikely, 2 • Unlikely, 3 • Somewhat unlikely, 4 • Somewhat likely, 5 • Likely, 6 • Very likely | 10 | .68 | .83 | .84 | .75 |
| Perceived vulnerability | Perceived risk at home: *'Do you think that anyone in your household is likely to be seriously ill if infected with the coronavirus'* 0—This question does not apply to me, 1 • Very unlikely, 2 • Unlikely, 3 • Somewhat unlikely, 4 • Somewhat likely, 5 • Likely, 6 • Very likely | a | .72 | .77 | .77 | .82 |
| Perceived vulnerability | Perceived risk for oneself: *'Do you think you are one of the people who could be seriously ill if infected by the coronavirus'* 0—This question does not apply to me, 1 • Very unlikely, 2 • Unlikely, 3 • Somewhat unlikely, 4 • Somewhat likely, 5 • Likely, 6 • Very likely | b | .51 | .68 | .62 | .75 |
| Threat-related emotions | Fear level during lockdown: *'Do you feel afraid during the lockdown?'* from not at all [1] to a lot [7] | c[2] | .86[3] | .89 | .93 | .76 |

*(Continued)*

**Table 1.** (Continued)

| Latent variable | Description | Item | Factor loading Sample 1W | Factor loading Sample 2 | Factor loading Sample 2M | Factor loading Sample 2W |
|---|---|---|---|---|---|---|
| Threat-related emotions | Anxiety level during lockdown: *'Do you feel anxious during the lockdown?'* from not at all [1] to a lot [7] | d[2] | -[3] | .64 | .59 | .81 |
| Perceived illness | One's probability of being infected: *'Do you think you have been infected with the coronavirus?'* 0—This question does not apply to me, 1 • Very unlikely, 2 • Unlikely, 3 • Somewhat unlikely, 4 • Somewhat likely, 5 • Likely, 6 • Very likely | e | 1[3] | .76 | .73 | .73 |
| Perceived illness | Probability of presenting symptoms: *'If you think you have been infected or are currently infected with the coronavirus, how do you rate your symptoms?'* 0—This question does not apply to me, 1 • Very unlikely, 2 • Unlikely, 3 • Somewhat unlikely, 4 • Somewhat likely, 5 • Likely, 6 • Very likely | f | -[3] | .77 | .80 | .83 |
| Lockdown behavior | *'Since the lockdown measure came into force, I have respected it and limited my outings'* from I disagree completely [1] to I agree completely [6] | g | .86 | .82 | .79 | .72 |
| Lockdown behavior | Number of unauthorized excursions: *'Since the lockdown measure came into force, I have limited my outings. I have gone on authorized outings, with a sworn statement'* from [1] outing to [5] outings (with increment of 1) | h | -.42 | -.53 | -.60 | -.41 |
| Lockdown behavior | Percent confined time: *'Since the lockdown measure came into effect, I have remained at home* from [0%] of the time to [100%] of the time (in increment of 20%) | i | .48 | .57 | .60 | .53 |
| Lockdown intentions | *'As long as the authorities require it, I will only leave home in case of extreme necessity and only for officially authorized reasons'* from I disagree completely [1] to I agree completely [6] | j | .91 | .93 | .92 | .93 |
| Lockdown intentions | *'I intend to respect the lockdown and therefore to limit my outings as much as possible as long as the authorities require it'* from I disagree completely [1] to I agree completely [6] | k | .93 | .93 | .94 | .91 |
| Lockdown intentions | *'I will respect the lockdown and therefore limit my outings as much as possible until the authorities decide otherwise'* from I disagree completely [1] to I agree completely [6] | l | .96 | .95 | .94 | .97 |
| Protective measures intentions | *'As long as the authorities demand it, I will respect all protective measures'* from I disagree completely [1] to I agree completely [6] | m | .99 | .97 | .97 | .98 |
| Protective measures intentions | *'I intend to respect all these protective measures as long as the authorities require it'* from I disagree completely [1] to I agree completely [6] | n | .96 | .94 | .95 | .93 |
| Protective measures intentions | *'I will respect these protective measures until the authorities decide otherwise'* from I disagree completely [1] to I agree completely [6] | o | .67 | .86 | .90 | .81 |

*(Continued)*

**Table 1.** (Continued)

| Latent variable | Description | Item | Factor loading Sample 1W | Factor loading Sample 2 | Factor loading Sample 2M | Factor loading Sample 2W |
|---|---|---|---|---|---|---|
| Protective measures behavior | '*Since protective measures came into effect, I have applied them on average…*' Never, 1–3 times a day, 4–6 times a day, 7–9 times a day, 10–12 times a day, All the time | p | .62 | .67 | .66 | .67 |
| Protective measures behavior | '*Since these protective measures have been in place, I have followed them all*' from I disagree completely [1] to I agree completely [6] | q | .90 | .79 | .84 | .71 |
| Protective measures behavior | '*Since these protective measures have been in place, I have followed them all*' from [0%] of the time to [100%] of the time (with increment of 20%) | r | .85 | .84 | .83 | .85 |
| Affiliation | '*During lockdown, time spent during the day chatting on social networks and/or watching, listening or reading about the coronavirus health crisis Social networks (Facebook, WhatsApp, Instagram, etc.)*' Never, at most 1 hour per day, about 2 hours per day, about 3 hours a day, about 4 hours a day, about 5 hours a day about 6 hours per day, about 7 hours a day or more | s | .52 | .69 | .74 | .64 |
| Affiliation | Time spent on the phone with close others: Same question structure and response choices as above, but with respect to '*Telephone conversations (with or without the image of your interlocutor on the screen)*' | t | .71 | .88 | .88 | .91 |
| Affiliation | Time spent on internet communication media (Skype etc.) with close others: Same question structure and response choices as above, but with respect to '*Internet conversations (Skype, Google-Hangout; Scopia, RenaVisio, etc.)*' | u | .62 | .84 | .90 | .74 |

*Note.* W = women; M = men.

[1]The two dimensions of SCO ('ability' and 'opinion') loaded on a second-order latent factor -global SCO- (see Figs 2 to 5) respectively at .75 and .86 for Sample 1W –women only, .87 and .87 for Sample 2, .82 and .91 for Sample 2W –women only–and .86 and .88 for Sample 2M –men only. Items 5, 7 and 11 of the SCO scale were removed due to low factor loadings in at least two models (see S1 Text for more details).

[2]The covariates for items 'fear' (c) and 'anxiety' (d) during lockdown, 'fear' (c') and 'anxiety' (d') before lockdown are not associated with factor loadings since they do not load on the Threat-Related Emotion factor. They are nevertheless presented in Figs 2 to 5.

[3] The items 'anxiety' (c) and 'probability of presenting symptoms' (f) were excluded from model Sample 1W because they produced factor loadings > 1 (see S1 Text for more details). The factors Threat-Related Emotions and Perceived Illness were estimated with one item each for that model only. Importantly, the structure of relations in Sample 1W between Threat-Related Emotions, Perceived Illness and Affiliation were largely replicated in Sample 2W with the removed items (see below), ruling out the possibility that differences in the hypothesized relations are due to the missing items.

noted that the threat-related emotions of fear and anxiety were jointly assessed and loaded on a single latent construct, since these emotions were specifically evoked during the COVID-19 pandemic [34] and exhibited very similar progression trends over a one-year period [52,53].

Figs 1 and 2, respectively, depict the hypothesized structural relationships (path coefficients) among constructs. A detailed description of the statistical procedure is available in S1 Text, which also reports fit indices for all models.

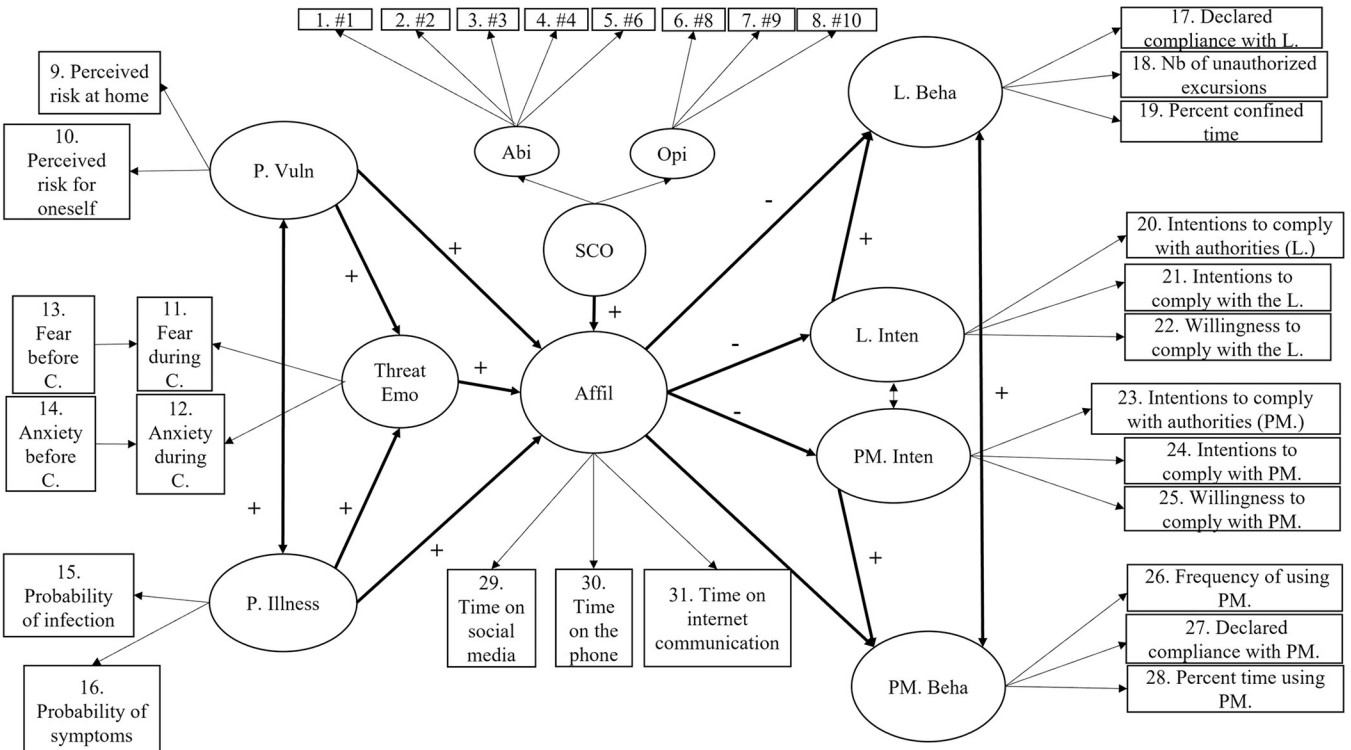

**Fig 1. Hypothesized direct relations between perceived vulnerability, perceived illness, threat-related emotions, social comparison, affiliation, intentions, and self-reported behavior regarding compliance with the lockdown and protective measures.** Note. The bold black lines represent expected direct relations (regressions and correlations). P.Vuln = perceived vulnerability; P. Illness = perceived illness; Threat Emo = threat-related emotions; Affil = affiliation; SCO = social comparison; Abi = SCO ability dimension; Opi = SCO opinion dimension; # = original item numbers of the SCO scale; L. Beha = behavioral self-reported compliance with the lockdown; L. Inten = Intentions to comply with the lockdown; PM. Inten = intentions to comply with the use of protective measures; PM. Beha = behavioral self-reported compliance with the use of protective measures.

## Mediation analyses

We performed mediation analyses [54] to examine the psychological processes underlying the antecedents and consequences of affiliative tendencies. According to Hayes, mediation analyses 'quantify and examine the direct and indirect pathways through which an antecedent variable X transmits its effect on a consequent variable Y through an intermediary M'. Irrespective of the current debate concerning the criteria used to establish mediation [54,55], we here consider Hayes' statement that 'it is the test of the indirect effect that matters, not the test on the individual paths in the model' [54] p. 125). We estimated four possible indirect effects (in bold pink in Fig 2), each corresponding to the product of the regression coefficients linking Y → M (the 'a' path) and M → X (the 'b' path). Two indirect effects focused on antecedents involving the influence of Perceived Vulnerability (X1) and Perceived Illness (X2) on Affiliation (Y) through Threat-Related Emotions (M). The coefficients for the indirect effects involving Perceived Vulnerability (X1) → Threat-Related Emotions (M) → Affiliation (Y), and Perceived Illness (X2) → Threat-Related Emotions (M) → Affiliation (Y) are referred to as βab1 and βab2. In turn, two indirect effects focused on consequences involving the influence of Affiliation (X) on Lockdown Behavior (Y1) and Protective Measures Behavior (Y2), through the associated respective compliance intentions (M1 and M2). The coefficients for the indirect effects involving Affiliation (X) → Lockdown Intentions (M1) → Lockdown Behavior (Y1), and Affiliation (X) → Protective Measures Intentions (M2) → Protective Measures Behavior

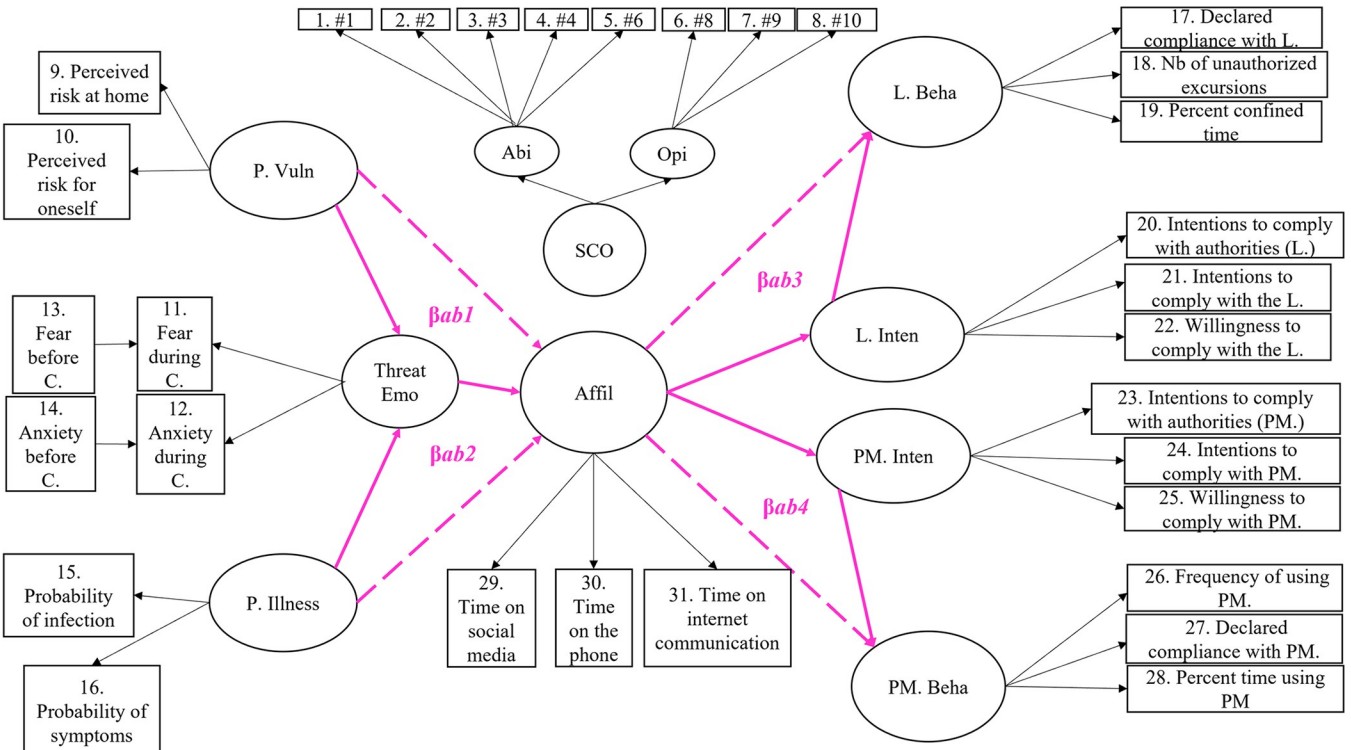

**Fig 2. The four hypothesized indirect relations.** *Note.* The bold pink lines and indices represent expected indirect relations (mediation effects). P. Vuln = perceived vulnerability; P. Illness = perceived illness; Threat Emo = threat-related emotions; Affil = affiliation; SCO = social comparison; Abi = SCO ability dimension; Opi = SCO opinion dimension; # = original item numbers of the SCO scale; L. Beha = behavioral self-reported compliance with the lockdown; L. Inten = Intentions to comply with the lockdown; PM. Inten = intentions to comply with the use of protective measures; PM. Beha = behavioral self-reported compliance with the use of protective measures. βab1 = Threat-related emotions' indirect effect on the prediction of affiliation by perceived vulnerability. βab2 = Threat-related emotions' indirect effect on the prediction of affiliation by perceived illness; βab3 = the indirect effect of intentions on the prediction of behavioral compliance with the lockdown by affiliation. βab4 = the indirect effect of intentions on the prediction of behavioral compliance with the protective measures by affiliation.

(Y2) are referred to as βab3 and βab4. In line with Hayes [54], and in order to establish mediation, we calculated and tested the statistical significance of each indirect effect (βab1, βab2, βab3 and βab4) and examined their respective amount of explained variance as a proportion of each total effect (direct effect + indirect effect).

## Comparisons of structural relations as a function of gender

We further examined whether the identified pattern of relationships varied depending on gender. We evaluated *gender-related variations in standardized latent means* to estimate the extent to which women and men varied in terms of the levels of the psychological constructs predicting compliance with pandemic rules (e.g., levels of affiliation tendencies and compliance with lockdown). We also assessed *gender-related variations in path coefficients* (i.e., regression and correlation coefficients) to evaluate the extent to which the magnitude of associations between the psychological constructs differed between women and men (e.g., whether men's levels of compliance with safety measures were influenced more strongly by their affiliation tendencies than women's). This was done using the multi-group measurement invariance procedure [56,57]. This technique makes it possible to examine group-related differences in latent variables, while ensuring that the scales reflect the constructs equally well across both women and men. There were three requirements: configural invariance (i.e., the hypothesized model had

the same structure across groups), metric invariance (i.e., the contribution of a given observed variable to its respective construct was equivalent across groups), and scalar invariance (i.e., in addition to previous assumptions, the intercept of a given observed variable was equal across groups).

This multi-group measurement invariance procedure was performed in the Lavaan package [50] version 0.6–3, and was conducted with sample 2, which offered a well-balanced proportion of women and men. Latent means and path coefficients of Sample 2W –women only– were compared to those of Sample 2M –men only– under scalar invariance, as recommended by Putnick and Bornstein [56]. Note that all full structural models showed a good fit according to current recommendations (see [58] and S1 Text).

## Results

Descriptive statistics for all study variables are available in S1 Table. Pearson correlations among all study variables for Samples 1W, 2M and 2W are available in in S2–S4 Tables. A false discovery rate correction for multiple significance testing using the Benjamini and Hochberg [59] method was applied when necessary. Given the large number of observed variables, all significance tests involving cross-sample mean differences and significance of correlation coefficients employed a false discovery rate (FDR) correction for multiple significance testing using the Benjamini and Hochberg method. In the case of structural models with latent variables, the FDR correction was applied only for significance tests related to cross-sample comparisons of latent means and regression coefficients.

### Sample 1W (women only)

We first examined the results of our model using Sample 1W (women only, $N = 827$). The 'anxiety' and 'probability of symptoms' items were excluded from model Sample 1W as they produced factor loadings > 1. Consequently, the latent variables Threat-Related Emotions and Perceived Illness were estimated with one item each for that model only. Importantly, the structure of relations in Sample 1W between Threat-Related Emotions, Perceived Illness, and Affiliation was largely replicated in Sample 2W with the removed items (see below), ruling out the possibility that differences in the hypothesized relations may have been due to the missing items.

As reported in Fig 3, a higher level of Perceived Vulnerability was associated with higher Threat-Related Emotions levels ($\beta = .24$, $p < .001$), while a higher level of Perceived Illness was not ($\beta = .06$, $p = .089$). These two predictors were unrelated ($r = .06$, $p = .170$). As expected, higher levels of SCO ($\beta = .15$, $p = .016$), Threat-Related Emotions ($\beta = .09$, $p = .047$) and Perceived Illness ($\beta = .09$, $p = .029$) were associated with greater Affiliation tendencies. Affiliation was unrelated to Perceived Vulnerability ($\beta = .06$, $p = .327$). In turn, Affiliation failed to predict Lockdown Intentions ($\beta = .02$, $p = .601$) and Protective Measures Intentions ($\beta = .02$, $p = .676$). Not surprisingly, more compliant Lockdown Intentions and Protective Measures Intentions were associated with greater reported compliance behaviors (Lockdown Behaviors: $\beta = .83$, $p < .001$; Protective Measures Behaviors: $\beta = .62$, $p < .001$). As shown by the indirect effects' values ($\beta ab1$ to $\beta ab4$) noted at the bottom of Fig 3, no significant mediations were found ($\beta$s $\leq$ .02, $p$s $\geq$ .076). However, the indirect effect of Perceived Vulnerability on Affiliation mediated by Threat-Related Emotions ($\beta ab1$ value) was marginally significant $\beta = .02$, $p = .076$. If significant, this indirect effect would indicate that higher levels of Perceived Vulnerability fostered greater Affiliation tendencies because individuals' Threat-Related Emotions increased during lockdown.

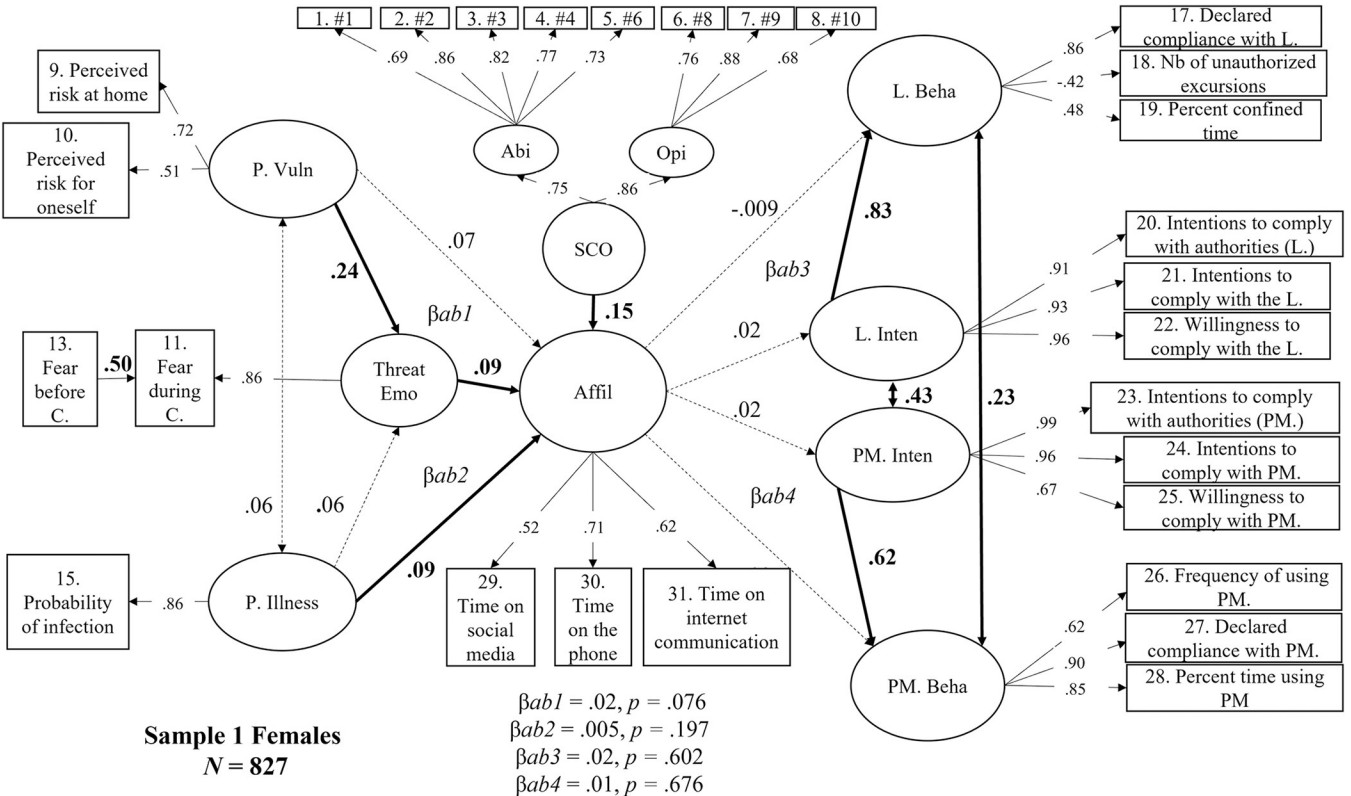

**Fig 3. Structural mediation model in women (Sample 1W, N = 827)** between perceived vulnerability, perceived illness, threat-related emotions, social comparison, affiliation, intentions, and self-reported behavior regarding compliance with the lockdown and protective measures. *Note.* Dotted lines represent non-significant effects at *p* < .05. Values in bold black and bold black lines represent significant direct effects or correlations. P.Vuln = perceived vulnerability; P. Illness = perceived illness; Threat Emo = threat-related emotions; Affil = affiliation; SCO = social comparison; Abi = SCO ability dimension; Opi = SCO opinion dimension; L. Beha = self-reported behavioral compliance with the lockdown; L. Inten = Intentions to comply with the lockdown; PM. Inten = intentions to comply with the use of protective measures; PM. Beha = behavioral self-reported compliance with the use of protective measures. β*ab1* = Threat-related emotions' indirect effect on the prediction of affiliation by perceived vulnerability; β*ab2* = Threat-related emotions' indirect effect on the prediction of affiliation by perceived illness; β*ab3* = the indirect effect of intentions on the prediction of behavioral compliance with the lockdown by affiliation. β*ab4* = the indirect effect of intentions on the prediction of behavioral compliance with the protective measures by affiliation.

To summarize, the more vulnerable the women in our sample perceived themselves as being to COVID-19, the more threat-related emotions they experienced and this, in turn, increased their affiliative tendencies. In addition, the more ill they perceived themselves to be and the more prone they were to compare themselves to similar individuals, the more affiliative tendencies they exhibited. Finally, women's intentions to comply with safety guidelines were associated with higher levels of reported compliance behaviors. However, there was no relationship between the extent to which they complied with safety guidelines and their affiliative tendencies.

## Sample 2 (men and women)

The measurement invariance procedure conducted with Lavaan [50] version 0.6–3 ensured that the scales in both groups presented equivalent psychometric properties (Table 2). Scalar invariance across gender groups was achieved in our model, with ΔCFI and ΔRMSEA values below the standard thresholds (ΔCFI = .01 and ΔRMSEA = .015, [60,61]. Deltas were below thresholds for the comparison of the configural and the metric invariance models and that of the metric and the scalar invariance models.

**Table 2. Measurement invariance analyses as a function of gender in Sample 2.**

| | Sample 2W vs Sample 2M | | | | |
|---|---|---|---|---|---|
| | $\chi^2$(df) | CFI | RMSEA | Δ CFI | Δ RMSEA |
| Configural | 1801.349 (832) | .952 | .047 | | |
| Metric | 1762.910 (852) | .949 | .048 | .003 | .001 |
| Scalar | 1907.586 (870) | .948 | .048 | .001 | .000 |

*Note.* W = women; M = men.

As summarized in Fig 4, we found that higher Perceived Vulnerability was associated with a higher level of Threat-Related Emotion (β = .33, $p < .001$), while Perceived Illness was not (-.07, $p = .089$). These two predictors correlated positively with each other ($r = .33$, $p < .001$). As a reminder, in Sample 1W, higher scores on SCO (β = .29, $p < .001$), Perceived Illness (β = .25, $p < .001$), and Threat-Related Emotions (β = .15, $p = .001$) were all associated with higher levels of Affiliation, whereas greater Perceived Vulnerability was not (β = -.07, $p = .098$).

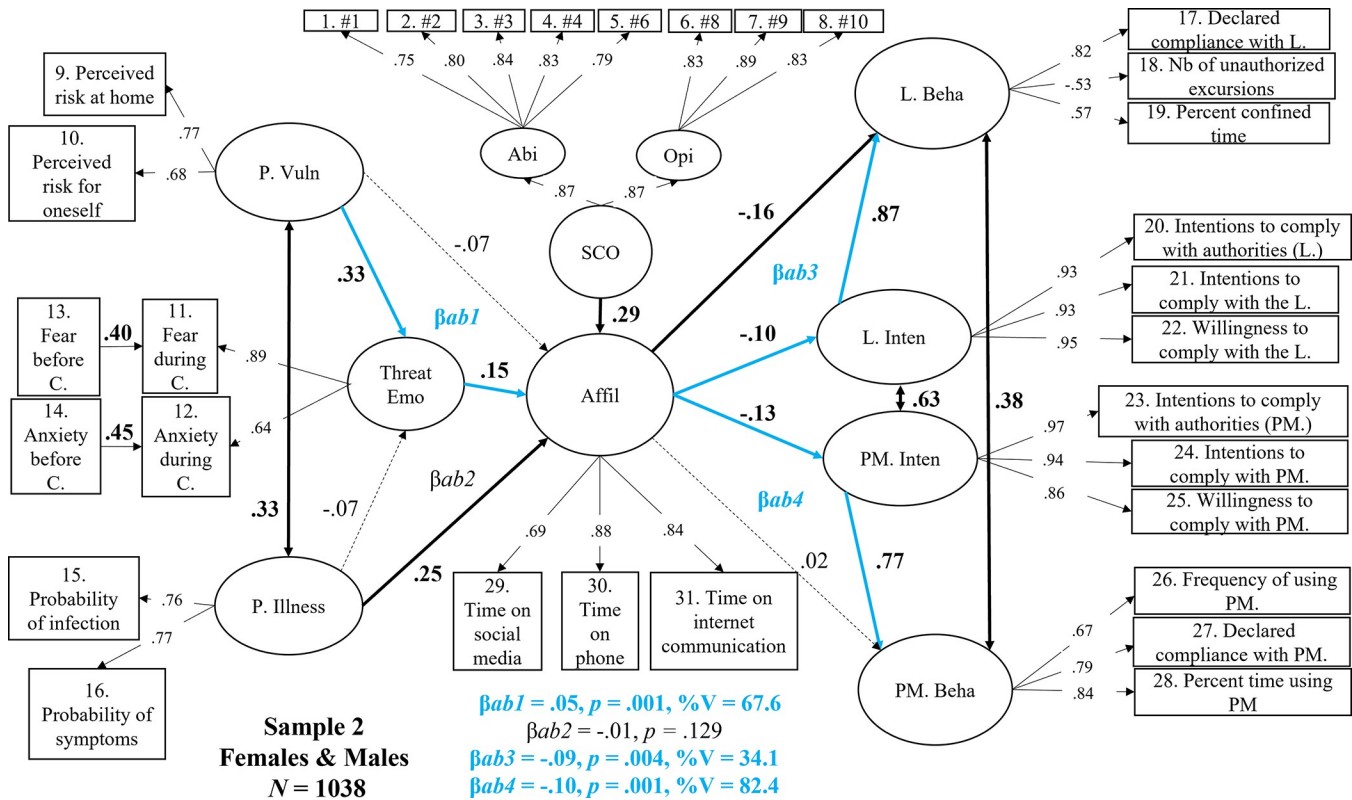

**Fig 4. Structural mediation model in women and men (Sample 2, N = 1038) between perceived vulnerability, perceived illness, threat-related emotions, social comparison, affiliation, intentions, and self-reported behavior regarding compliance with the lockdown and protective measures.** *Note.* Dotted lines represent non-significant effects at $p < .05$. Values in bold black and bold black lines represent significant direct effects or correlations. Blue lines and values highlight significant indirect (mediation) effects. P.Vuln = perceived vulnerability; P. Illness = perceived illness; Threat Emo = threat-related emotions; Affil = affiliation; SCO = social comparison; Abi = SCO ability dimension; Opi = SCO opinion dimension; L. Beha = behavioral self-reported compliance with the lockdown; L. Inten = Intentions to comply with the lockdown; PM. Inten = intentions to comply with the use of protective measures; PM. Beha = behavioral self-reported compliance with the use of protective measures. βab1 = Threat-related emotions' indirect effect on the prediction of affiliation by perceived vulnerability. βab2 = Threat-related emotions' indirect effect on the prediction of affiliation by perceived illness; βab3 = the indirect effect of intentions on the prediction of behavioral compliance with the lockdown by affiliation. βab4 = the indirect effect of intentions on the prediction of behavioral compliance with the protective measures by affiliation.

Importantly, and as expected, a higher Affiliation score predicted *less compliant* Lockdown Intentions (β = -.10, *p* = .004) as well as *less compliant* Protective Measures Intentions (β = -.13, *p* < .001). In turn, each type of intention (again correlated with each other) positively and strongly predicted the corresponding reported behavior (Lockdown: β = .87, *p* < .001; Protective measures: β = .77, *p* < .001). The mediation analyses revealed significant indirect effects (highlighted in blue in Fig 4). They showed that a higher level of Perceived Vulnerability predicted greater Affiliation tendencies only because individuals' Threat-Related Emotions increased during lockdown (β = .05, *p* = .001, an indirect effect explaining a large portion of variance– 67.6%). Analyses also revealed links between Affiliation, Intentions, and Behaviors. A higher level of Affiliation predicted less compliant Lockdown Behaviors, an effect that was partly accounted for (34.1%) by a lower level of compliant Lockdown Intentions (β = -.09, *p* = .004). The presence of a still significant direct effect (β = -.16, *p* < .001) means that affiliative tendencies that reduced lockdown compliance also occurred regardless of individuals' intentions to comply or not (in fact, most of the time– 65.9%). Furthermore, Affiliation predicted less compliant Protective Measures Behaviors, primarily due (at 82.4%) to the fact that individuals reported less compliant Protective Measures Intentions (β = -.10, *p* < .001) the more affiliated they were to those close to them.

To summarize the overall structural relations found in women and men in Sample 2, the more vulnerable to COVID-19 participants perceived themselves, the more threat-related emotions they experienced and this, in turn, increased their affiliative tendencies. In addition, the more ill participants perceived themselves, the more prone they were to compare themselves to similar individuals, the higher the level of affiliative tendencies they exhibited. Participants' intentions to comply with safety guidelines were associated with a higher level of reported compliance behaviors. Interestingly, when we consider women and men taken together, we find that the more they affiliated online, the less they tended to comply with safety guidelines, a finding that is consistent with Dezecache et al.'s [26] hypothesis.

## Sample 2M (focus on men participants)

We next examined Sample 2M (men only, Fig 5) and Sample 2W (women only, Fig 5) separately and discovered that the expected critical negative links between Affiliation and Compliance Intentions (and ultimately reported Behaviors) found with Sample 2 were actually valid for men but not for women (a finding consistent with the lack of these critical links in Sample 1W –men only).

In Sample 2M (men only), we found that greater Perceived Vulnerability was associated with higher levels of Threat-Related Emotion (β = .38, *p* < .001), whereas this relation was also significant but negative in the case of Perceived Illness (-.16, *p* = .008). These two predictors correlated positively with each other (*r* = .39, *p* < .001). As found in all previous models, higher SCO (β = .32, *p* < .001), Perceived Illness (β = .31, *p* < .001) and Threat-Related Emotions scores (β = .15, *p* = .006) were all associated with a higher level of Affiliation, whereas Perceived Vulnerability was not (β = -.11, *p* = .113).

Regarding the structural relations, higher Affiliation levels predicted *less compliant* Lockdown Intentions (β = -.11, *p* = .021) as well as *less compliant* Protective Measures Intentions (β = -.19, *p* < .001). In turn, each type of intention (clearly correlated with each other) positively and strongly predicted the corresponding reported behavior (Lockdown Behavior: β = .84, *p* < .001; Protective Measures Behavior: β = .81, *p* < .001). In the same way as for the overall Sample 2, the mediation analyses revealed that higher levels of Perceived Vulnerability fostered greater Affiliation tendencies only because individuals' Threat-Related Emotions increased during lockdown (β = .06, *p* = .020, an indirect effect explaining 52.3% of variance). It should be noted that the counterpart to this effect, with Perceived Illness as the predictor, was

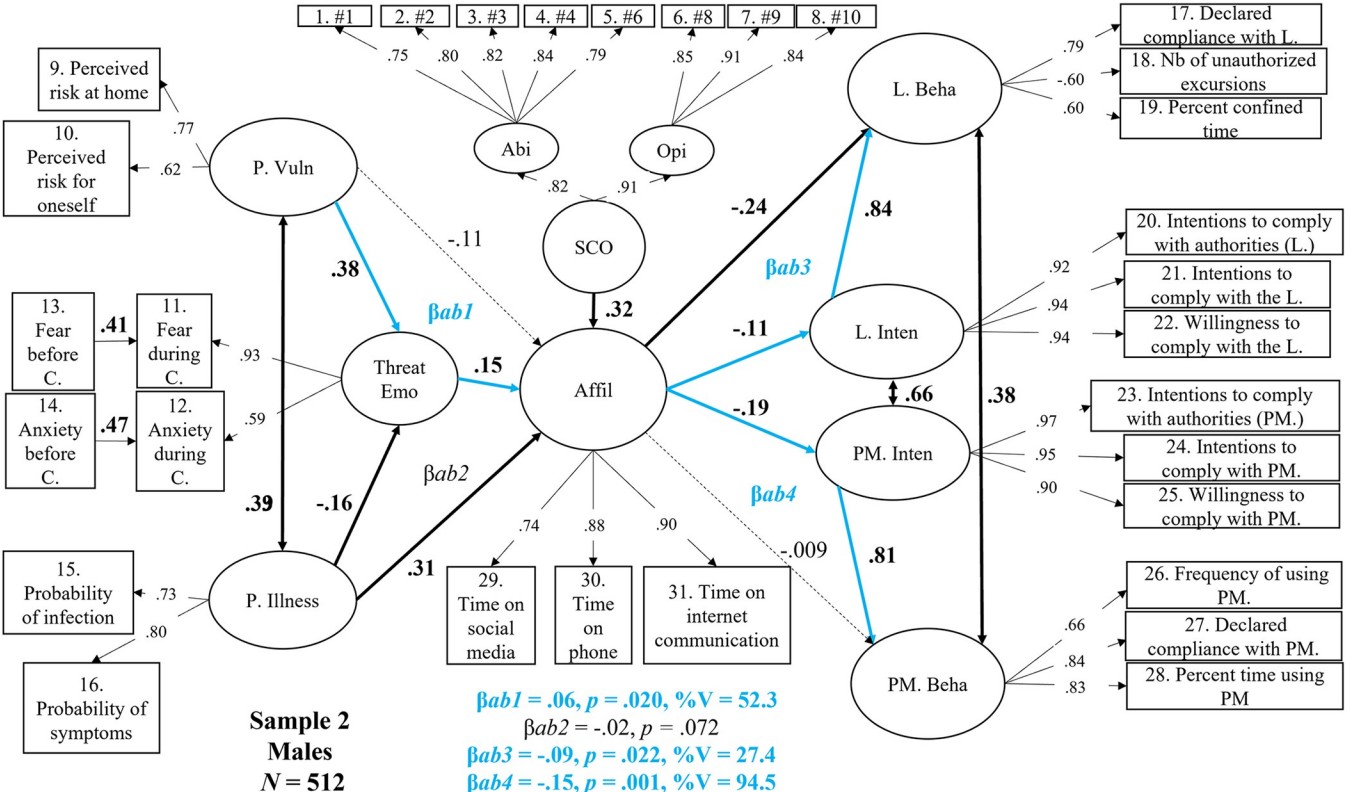

**Fig 5. Structural mediation model in men (Sample 2M, N = 512) between perceived vulnerability, perceived illness, threat-related emotions, social comparison, affiliation, intentions, and self-reported behavior regarding compliance with the lockdown and protective measures.** *Note.* Dotted lines represent non-significant effects at *p* < .05. Values in bold black and bold black lines represent significant direct effects or correlations. Blue lines and values highlight significant indirect (mediation) effects. P.Vuln = perceived vulnerability; P. Illness = perceived illness; Threat Emo = threat-related emotions; Affil = affiliation; SCO = social comparison; Abi = SCO ability dimension; Opi = SCO opinion dimension; L. Beha = behavioral self-reported compliance with the lockdown; L. Inten = Intentions to comply with the lockdown; PM. Inten = intentions to comply with the use of protective measures; PM. Beha = behavioral self-reported compliance with the use of protective measures. βab1 = Threat-related emotions' indirect effect on the prediction of affiliation by perceived vulnerability. βab2 = Threat-related emotions' indirect effect on the prediction of affiliation by perceived illness; βab3 = the indirect effect of intentions on the prediction of behavioral compliance with the lockdown by affiliation. βab4 = the indirect effect of intentions on the prediction of behavioral compliance with the protective measures by affiliation.

marginally significant (β = .02, *p* = .072). Links were also found between Affiliation, Intentions, and Behaviors. Higher Affiliation levels predicted less compliant Lockdown Behaviors, an effect that was partly accounted for (27.4%) by less compliant Lockdown Intentions (β = -.09, *p* = .022). The presence of a still significant and larger direct effect (β = -.24, *p* < .001) means that affiliative tendencies that reduced lockdown compliance generally occurred regardless of individuals' intentions to comply or not (72.6%). Furthermore, Affiliation predicted less compliant Protective Measures Behaviors, explained to a very large extent (at 94.5%) by the fact that individuals' Protective Measures Intentions were less compliant (β = -.15, *p* < .001) the more they affiliated with their close ones.

The structural relations found specifically for men in Sample 2 were very similar to those observed in Sample 2 as a whole. Men's tendency to affiliate online was influenced by their inclination to compare themselves to similar individuals as well by their perceptions of vulnerability resulting from their higher levels of experience of threat-related emotions. More importantly, the reduction in compliance with safety guidelines in men was associated with a greater tendency to affiliate online. We now examine whether these critical relations were also observed in women.

Table 3. Standardized latent mean comparisons between Sample 2W and Sample 2M (gender effects).

| Factors | $\Delta$ Sample 2M | $p^1$ |
|---|---|---|
| Perceived Vulnerability | -.08 | .296 |
| Perceived Illness | .14 | .068 |
| Threat-Related Emotions | -.28 | **= .007** |
| SCO | .08 | .09 |
| Ability | .04 | .186 |
| Opinion | .002 | .954 |
| Affiliation | .14 | .056 |
| Lockdown Intentions | -.25 | **< .001** |
| Protective Measures Intentions | -.37 | **< .001** |
| Lockdown Behavior | -.15 | **= .010** |
| Protective Measures Behavior | -.07 | .233 |

*Note*: $\Delta$ Sample 2M = Standardized Mean differences Sample 2M –Sample 2W.

[1]All p values in bold survived the FDR correction for multiple comparisons.

## Sample 2W (focus on women participants)

As reported in Fig 6, Sample 2W (women only) offered a different picture regarding the critical links between affiliation and compliance. As suggested earlier, none of these links were significant even though the statistical power was similar to that for Sample 2M. Furthermore, the findings for Sample 2W largely replicated the findings of Sample 1W.

We found that greater Perceived Vulnerability was associated with higher levels of Threat-Related Emotion ($\beta$ = .29, $p < .001$), while Perceived Illness was not (.06, $p = .275$). These two predictors positively correlated with each other ($r = .25$, $p < .001$). As found in all previous models, higher SCO ($\beta$ = .25, $p < .001$), Perceived Illness ($\beta$ = .17, $p = .004$) and Threat-Related Emotions scores ($\beta$ = .21, $p < .001$) were all associated with higher Affiliation levels, whereas Perceived Vulnerability was not ($\beta$ = -.06, $p = .275$).

In line with Sample 1W but by contrast to Sample 2M, higher levels of Affiliation were unrelated to Lockdown Intentions ($\beta$ = -.07, $p = .149$) and Protective Measures Intentions ($\beta$ = -.04, $p = .411$). Each type of intention (correlated with each other) positively and strongly predicted the corresponding reported behavior (Lockdown Behavior: $\beta$ = .88, $p < .001$; Protective Measures Behavior: $\beta$ = .69, $p < .001$). The mediation analyses revealed an indirect effect that was only marginally significant in Sample 1W but significant in all the other models, namely that higher levels of Perceived Vulnerability fostered greater Affiliation tendencies as individuals' Threat-Related Emotions increased during lockdown ($\beta$ = .06, $p = .004$, up to 48.4% of variance explained). No other indirect effects were significant ($\beta$ = -.06, $p = .149$), a finding consistent with Sample 1W but different from Sample 2M.

The structural relations found in Sample 2 for women show that the tendency to affiliate online in times of pandemic is shared by women and men: they affiliate more when they experience more intense threat-related emotions resulting from greater perceived vulnerability. Second, the reduction in compliance with safety guidelines associated with greater tendencies to affiliate online is specific to men.

## Gender-related variations in standardized latent means, regression, and correlation coefficients

To gain a more in-depth understanding of the gender differences in Sample 2, we further examined the structural differences by directly comparing women and men. While the

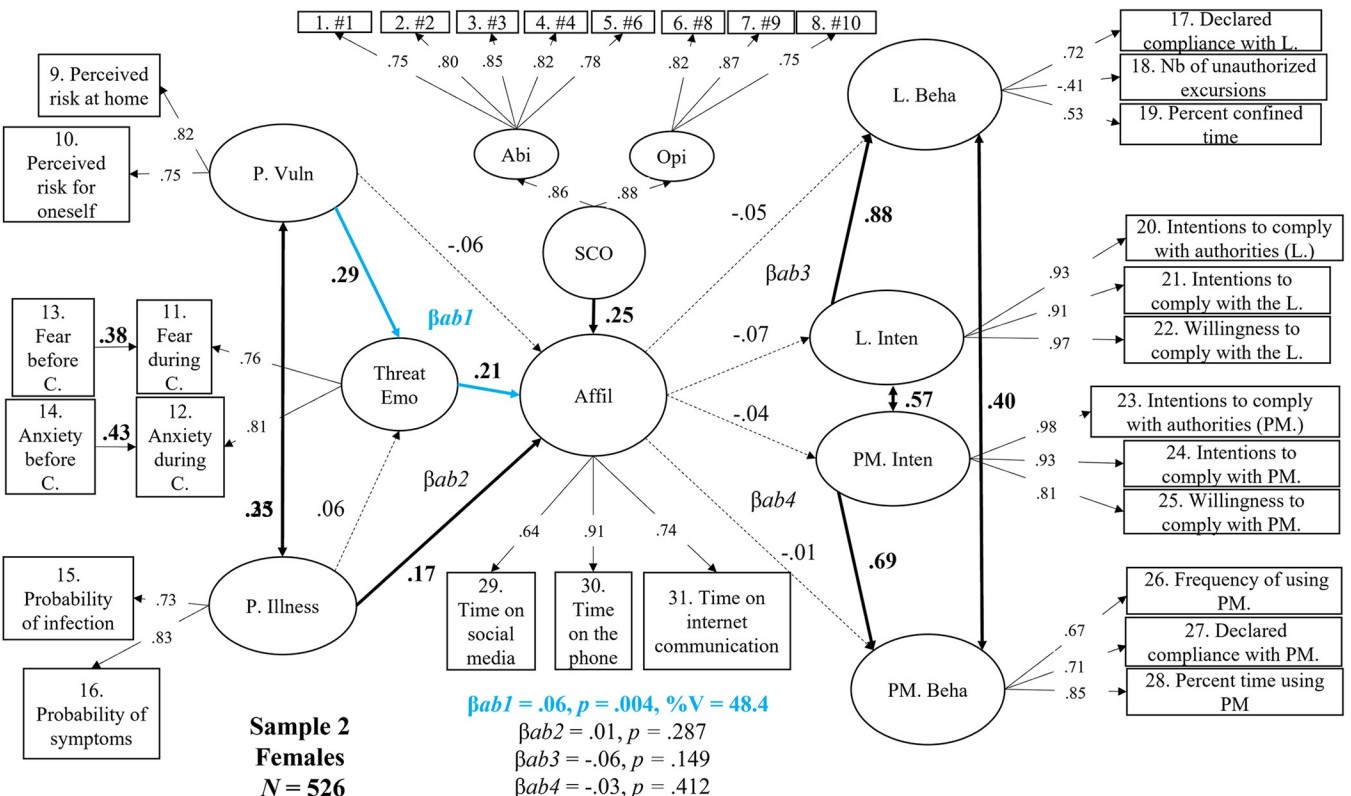

**Fig 6. Structural mediation model in women (Sample 2W, N = 526) between perceived vulnerability, perceived illness, threat-related emotions, social comparison, affiliation, intentions, and self-reported behavior regarding compliance with the lockdown and the protective measures.** *Note*. Dotted lines represent non-significant effects at *p* < .05. Values in bold black and bold black lines represent significant direct effects or correlations. Blue lines and values highlight significant indirect (mediation) effects. P.Vuln = perceived vulnerability; P. Illness = perceived illness; Threat Emo = threat-related emotions; Affil = affiliation; SCO = social comparison; Abi = SCO ability dimension; Opi = SCO opinion dimension; L. Beha = behavioral self-reported compliance with the lockdown; L. Inten = Intentions to comply with the lockdown; PM. Inten = intentions to comply with the use of protective measures; PM. Beha = behavioral self-reported compliance with the use of protective measures. βab1 = Threat-related emotions' indirect effect on the prediction of affiliation by perceived vulnerability. βab2 = Threat-related emotions' indirect effect on the prediction of affiliation by perceived illness; βab3 = the indirect effect of intentions on the prediction of behavioral compliance with the lockdown by affiliation. βab4 = the indirect effect of intentions on the prediction of behavioral compliance with the protective measures by affiliation.

measurement invariance results shown in Table 2 suggest that constructs were indeed interpreted in the same way by both gender groups, group differences in standardized values may have existed at the latent level, including differences in latent means and the magnitude of path coefficients. These differences can be compared after establishing scalar invariance (i.e., gender-related variations at the construct level are therefore not attributable to differences in intercepts and factor loadings across groups [56,62]).

Latent mean differences (delta values) were computed under scalar invariance [56]) using the Lavaan package [50] version 0.6–3. An FDR correction for multiple comparisons was applied to the latent mean comparisons using the Benjamini-Hochberg method [59]. As shown in Table 3, we found that men exhibited significantly lower levels of Threat-Related Emotions (Δ = -.28, *p* = 007), compliant Lockdown Intentions (Δ = -.25, *p* < 001), Protective Measures Intentions (Δ = -.37, *p* < 001) and compliant Lockdown Behavior (Δ = -.15, *p* = 010) than women.

We also compared regression (direct and indirect relations) and correlation coefficients across gender groups using a series of Wald tests [62,63]. The procedure was performed under scalar invariance and an FDR correction for multiple comparisons was applied. As shown in

**Table 4. Regressions and correlation comparisons between Samples 2W and 2M.**

| | | β Sample 2W (N = 526) | β Sample 2M (N = 512) | W(1) | p[1] |
|---|---|---|---|---|---|
| *Regressions (direct relations)* | | | | | |
| Fear before lockdown | Fear during lockdown | .38*** | .41*** | .27 | .60 |
| Anxiety before lockdown | Anxiety during lockdown | .43*** | .47*** | .56 | .47 |
| Perceived Vulnerability | Threat-Related Emotions | .27*** | .40*** | 1.84 | .17 |
| Perceived Illness | Threat-Related Emotions | .03 | -.16* | 4.82 | **= .028** |
| Threat-Related Emotions | Affiliation | .18** | .17** | .30 | .58 |
| SCO | Affiliation | .27*** | .32*** | 3.44 | .063 |
| Affiliation | Lockdown Intentions | -.09 | -.10* | .001 | .97 |
| Affiliation | Protective Measures Intentions | -.05 | -.18*** | 3.59 | .057 |
| Affiliation | Lockdown Behavior | -.08* | -.21*** | 6.55 | **= .010** |
| Affiliation | Protective Measures Behavior | -.02 | -.01 | .04 | .84 |
| Lockdown Intentions | Lockdown Behavior | .89*** | .84*** | 1.66 | .20 |
| Protective Measures Intentions | Protected Measures Behavior | .71*** | .81*** | .93 | .33 |
| *Mediations (indirect relations)* | | | | | |
| β*ab1* Perceived Vulnerability → Threat-Related Emotions → Affiliation | | .05** | .06* | 1.10 | .30 |
| β*ab2* Perceived Illness → Threat-Related Emotions → Affiliation | | .005 | -.03 | 3.46 | .062 |
| β*ab3* Affiliation → Lockdown Intentions → Lockdown Behavior | | -.08 | -.08* | .02 | .89 |
| β*ab4* Affiliation → Protective Measures Intentions → Protective Measures Behavior | | -.04 | -.14*** | 3.96 | **= .046** |
| *Correlations* | | | | | |
| Perceived Vulnerability | Perceived Illness | .25*** | .41*** | 3.52 | .060 |
| Protective Measures Intentions | Lockdown Intentions | .57*** | .66*** | 15.08 | **< .001** |
| Lockdown Behavior | Protective Measures Behavior | .43*** | .36*** | .13 | .72 |

*Note*. The reported coefficient varies slightly between Figs 3 and 4 and the present table because scalar invariance caused factor loadings and intercepts to be equal across groups. W = women; M = men

* = significant at the 0.05 level

** = significant at the 0.01 level

*** = significant at the 0.001 level.

1 All p values in bold survived the FDR correction for multiple comparisons.

Table 4, men and women differed on the following relations: by contrast to women (β = .03, *p* = .562), the more the men perceived themselves to be ill, the lower the level of Threat-Related Emotions they experienced (β = -.06, *p* = .012), W = 4.82, p = .03. Men exhibited a larger negative influence of Affiliation than women (β = -.08, *p* = .038) and this resulted in less compliant Lockdown Behavior (β = .21, *p* < .001), W = 6.55, p = .010. The mediating power of Protective Measures Intentions on Protective Measures Behavior was greater in men (β = -.14, *p* < .001) than in women (β = -.04, *p* < .290). Finally, the correlation coefficient between Protective Measures Intentions and Lockdown Intentions was larger in men (β = .66, *p* < .001) than in women (β = .57, *p* < .001), W = 15.08, *p* < .001. In sum, affiliation tendencies caused men to perceive less threat and comply less with the guidelines than women.

## Discussion

In this research, we looked at whether basic affiliative needs (measured here on the basis of reported online affiliative behavior) might have reduced adherence to COVID-19 health

guidelines, in line with the hypothesis outlined in [26]. The study was conducted using two samples of French participants recruited during the first lockdown in France (March-May 2020). Our first sample consisted primarily of women and was recruited via social networks. This gender imbalance was corrected in a second sample with a balanced gender composition. We have reported our analyses of both samples to ensure transparency and make it possible to evaluate the replicability of our findings.

Taken together, our results reveal that the expected link (greater affiliation tendency results in lower compliance) only holds for men, with women reporting greater adherence to health guidelines as well as other differences. This pattern (which was not hypothesized by [26]) should not be surprising considering the wealth of data showing that women, ceteris paribus, considered the virus to be a major health issue [31] and felt more responsible for ensuring collective protection [64], even though they were themselves at less risk of suffering from COVID-19-related health outcomes ([65]).

In Sample 1 specifically (women only), our mediation analyses do not reveal a significant indirect effect of Perceived vulnerability on Affiliation mediated by Threat-related emotions. Perceived vulnerability predicted changes in Threat-related emotions. However, the relationship between Perceived illness and Threat-related emotions was not significant. It is possible that, in this sample, the idea of being infected by COVID-19 (one's perceived illness) did not elicit threat-related emotions given that testing was not readily available at the time, making infection difficult to imagine and symptoms difficult to ascertain. It is also possible that the understanding of infections is based on intuitive theories [66], with the emergence of COVID-19 not being well enough understood to permit a clear assessment of one's own infectious state. As predicted based on the link between SCO and interpersonal orientation [67], scores on the SCO positively influenced affiliation, measured as time spent on social and other online or virtual (inc. telephone) media. We also found that threat-related emotions (anxiety and fear) and Perceived illness influenced affiliation activity, a finding that is consistent with a considerable body of literature indicating that affiliative behavior [24,25] reduces stress and/or uncertainty in threatening contexts [20]. However, affiliation failed to predict reported intentions to abide by lockdown and protection guidelines in this sample (mostly composed of women), contrary to the hypothesis of Dezecache et al. [26].

The results of Sample 2 show the indirect effect of Perceived vulnerability on Affiliation mediated by Threat-related emotions. In addition, and in line with our main hypothesis, they show the indirect effect of Affiliation on behavior mediated by intentions to comply both with lockdown and protective measures. As in Sample 1, Affiliation was positively affected by SCO. Again, as in Sample 1, Perceived illness did not influence Affiliation through Threat-related emotions, although we can note that this measure was positively related to Affiliation. One possibility is that the feeling of infectability drives affiliative needs and causes people to seek companionship when they feel they are at risk of being infected without, however, this being accompanied by a feeling of fear and/or anxiety. This would be contrary to the spontaneous behavior displayed by a range of species that seek social isolation or reduced social behavior when infected [68–70]. It is, however, possible that, as in other social species (e.g., [71]), dispositions have evolved to detect and selectively avoid infected conspecifics [72,73], thus helping to maintain social isolation despite the unwillingness of infected individuals to self-isolate. This disposition may also be supported by our known capacity to detect pathogen-bearing elements and avoid them due to the phenomenological experience of disgust [74,75]. Obviously, questionnaire items are related to the respondent's perception of the probability of being infected and it is likely that there will be discrepancies between theoretical responses and actual behavior when infection strikes. In this context, information from Google mobility data and

other similar sources could ultimately be used to assess 'actual behavior' during the lockdown [76].

Overall, the data from Sample 2, but not that from Sample 1, is consistent with Dezecache et al.'s [26] hypothesis. How do the samples differ? As we have discussed at length, these differences are explained by gender. If the mediation between Perceived vulnerability and Affiliation through Threat-related emotions is present regardless of whether men are excluded or included in Sample 2, the presence of men in this sample nevertheless explains the mediation between Affiliation and Behaviors (with respect to both Lockdown and Protective measures) via reported Intentions (again, with respect to both Lockdown and Protective measures). In other words, it is because Sample 2 contains a larger number of men that we observe the tendency to infringe guidelines.

The role of gender as a source of psychological variation and behavior has been widely debated, and particularly so during the COVID-19 pandemic, with research examining the role of gender in risk-taking [31]. These studies were based on the literature on gender differences in risk-taking [77] or the willingness to self-signal as a risk-taker [78], with risk-taking being likely to negatively influence compliance with safety guidelines [79]. Indeed, other differences associated with gender investigated in COVID-19 research (related to fear of the virus [35], coping style [39], readiness to endure sacrifices for the collective good [64] or other psychological traits such as agreeableness [76]) could well help explain greater adherence to guidelines. One intriguing possibility, which would also explain a (to our knowledge) as yet unexplored difference between women and men observed by Galasso and colleagues [31], is that ability to interact with others only online may not satisfy men's affiliative needs, leading them to break the rules in order to fulfill these needs. Irrespective of gender differences in risk-taking, all the participants in our samples were subject to strict police controls. Although Galasso et al. [31] suggest using gender-based public health communication, we believe one necessary step is to examine the different ways women and men used social media during the pandemic to fulfill their affiliative needs which, we believe, were heightened during this period in the same way they are during other disasters. Studies suggest that men and women use online social media in different ways, with women using them to maintain existing relations and men using them to meet new people [80]. Other studies suggest that women make greater use of online communication (e.g., [81].

Our results suggest that the COVID-19 "gender gap" is wider than previously believed, adding to the facts that women reported being willing to sacrifice more [64], being more likely to use scientific knowledge to inform their behavior [82], but also more likely to suffer the consequences of public health investment in COVID-19 treatment at the expense of maternity and infant care [83], increased domestic violence [84], and deteriorating mental health [85].

We see several limitations to our work:

Firstly, and as explained in the Introduction, it not entirely clear whether our data was sourced based on 'self-defined' gender. The various works on so-called 'gender' differences in COVID-19 might be addressing what should more properly be termed 'sex'. We must remain neutral on this question and are unable to examine it further because we did not request this specific information from our participants. Because the behaviors we measured (self-reported psychological state and behavior) are extremely susceptible to social influence, it is very likely that our results reflect 'socialization practices' and/or 'social roles' rather than the effects of biology (being born and growing up as a male or female, with distinctive biological profiles).

Second, important demographic information is lacking. Demographic elements that were not collected (besides gender) could explain the observed difference better than our subdivision into two genders. For instance, we did not collect race/ethnicity data because to do so is against the law of the country of study. We also had no precise information on geographical

location, which could have been critical (there are known differences in well-being depending on place of residence during the lockdown (e.g., [86]). Finally, we have only limited measures of economic insecurity, which is otherwise known to have played a role in COVID-19-related transgressive behaviors [87].

Third, the quality of sampling in Sample 1 (notably related to 'lack of representativeness' and 'self-selection') is questionable because data collection took place via social media. Sample 2, however, is a representative sample. This does not eliminate 'self-selection' issues. These can never be eliminated given the ethical constraints of psychological research, which require participants to give their voluntary consent.

Fourth, we did not measure 'affiliation' *per se*, but a rough index of 'affiliative need' via investment in online communication. We were not able to measure 'affiliation behavior' based on visits outside the household because to do so would have introduced confusion between our indices of 'affiliation' and 'transgression'. We could not measure affiliative needs directly because our research is based on secondary data analysis [52,53].

Fifth, it is possible that self-reports are influenced by gender with, e.g., women being more fearful of the potential consequences of revealing non-compliant behaviors. This is unlikely given that participants responded anonymously to our questionnaire and that it was not completed in the presence of the data collectors and/or employees of the market research company. Also, the fear of reporting non-compliant behavior would not explain other gender differences observed in this study. Additionally, if we were to pay heed to this potential objection, we would also have to eliminate all research efforts aimed at uncovering gender differences in other contexts. Participants are always able to lie and/or omit information, and there is no way to capture 'true' behavior apart from direct observation (which was not possible since excursions away from home were restricted in France).

Sixth and finally, our data were collected in the early days of the pandemic, and it is unclear how our findings would replicate during its subsequent stages, given the swift evolution of societal responses and adaptation. Given that the main hypothesis we tested were predicated upon feelings of uncertainty and fear (and the fact that lockdown was imposed), our results could help to predict (albeit probably only to some limited degree) non-compliance resulting from increased affiliative needs in cases where levels of fear of the virus are already high.

Despite this and other methodological weaknesses associated with the use of questionnaires (which make use of self-reported measures and not actual behavior), our results are compatible with the idea that for men, the feeling of vulnerability to disease during the lockdown was associated with increased affiliative needs. These were mediated by an increase in self-assessed threat-related emotions. In men specifically, and in France at least, greater affiliative needs could have created a need to infringe the public health and lockdown guidelines. This confirms that affiliation, and more generally social and group bonds [10,88], are strong determinants of how people behave in extreme circumstances, including in the case of a pandemic.

Affiliative needs, basic though they may seem to be, could be of great importance for the implementation of health guidelines. They should be given full consideration in future research and, ultimately, policy-making.

## Supporting information

**S1 Text. Detailed statistical procedure.**
(PDF)

**S1 Table. Descriptive statistics for all study variables in Samples 1W, 2M and 2W.**
(PDF)

**S2 Table. Pearson correlation matrix for all study variables in Sample 1W.**
(PDF)

**S3 Table. Pearson correlation matrix for all study variables in Sample 2M.**
(PDF)

**S4 Table. Pearson correlation matrix for all study variables in Sample 2W.**
(PDF)

## Author Contributions

**Conceptualization:** Guillaume Dezecache, Johann Chevalère, Natalia Martinelli, Sandrine Gil, Clément Belletier, Sylvie Droit-Volet, Pascal Huguet.

**Data curation:** Johann Chevalère, Natalia Martinelli, Sylvie Droit-Volet.

**Formal analysis:** Johann Chevalère.

**Funding acquisition:** Natalia Martinelli, Sylvie Droit-Volet, Pascal Huguet.

**Investigation:** Johann Chevalère, Natalia Martinelli, Sylvie Droit-Volet, Pascal Huguet.

**Methodology:** Johann Chevalère, Natalia Martinelli, Clément Belletier, Sylvie Droit-Volet, Pascal Huguet.

**Project administration:** Natalia Martinelli, Sandrine Gil, Sylvie Droit-Volet, Pascal Huguet.

**Resources:** Johann Chevalère.

**Software:** Johann Chevalère.

**Supervision:** Guillaume Dezecache, Sandrine Gil, Sylvie Droit-Volet, Pascal Huguet.

**Validation:** Guillaume Dezecache, Johann Chevalère.

**Visualization:** Johann Chevalère, Pascal Huguet.

**Writing – original draft:** Guillaume Dezecache, Johann Chevalère, Pascal Huguet.

**Writing – review & editing:** Guillaume Dezecache, Johann Chevalère, Natalia Martinelli, Sandrine Gil, Clément Belletier, Sylvie Droit-Volet, Pascal Huguet.

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
