## [Decision Letter · Decision Letter 0]

31 May 2024

PONE-D-24-10529Affiliation in times of pandemics: Determinants and consequencesPLOS ONE

Dear Dr. Dezecache,

Thank you for submitting your manuscript to PLOS ONE. After careful consideration, we feel that it has merit but does not fully meet PLOS ONE’s publication criteria as it currently stands. Therefore, we invite you to submit a revised version of the manuscript that addresses the points raised during the review process.

The reviewers and I all agree that the manuscript is greatly improved. Reviewer 2 points out a few minor issues that require revision, but these should be easy enough to address in a revised manuscript. I am formally recommending “Minor Revisions” to indicate that these revisions must be done, but I’d like to also note that if they addressed satisfactorily then the manuscript should be publishable. I look forward to reading the revised manuscript.

We look forward to receiving your revised manuscript.

Kind regards,

Pat Barclay

Academic Editor

PLOS ONE

2. Thank you for stating the following financial disclosure: "This work was supported by the French ANR (ANR-Flash COVID-19)." 

3. Please expand the acronym “ANR” (as indicated in your financial disclosure) so that it states the name of your funders in full.

Reviewers' comments:

Reviewer's Responses to Questions

**Comments to the Author**

1. Is the manuscript technically sound, and do the data support the conclusions?

Reviewer #1: Yes

Reviewer #2: Yes

2. Has the statistical analysis been performed appropriately and rigorously? 

Reviewer #1: Yes

Reviewer #2: Yes

3. Have the authors made all data underlying the findings in their manuscript fully available?

Reviewer #1: Yes

Reviewer #2: Yes

4. Is the manuscript presented in an intelligible fashion and written in standard English?

Reviewer #1: Yes

Reviewer #2: Yes

5. Review Comments to the Author

Reviewer #1: I have been asked to treat this submission as a resubmission of a previous submission (manuscript ID PONE-D-22-34262). The comments that I raised in that previous version have been satisfactorily addressed.

Reviewer #2: Review of “Affiliation in times of pandemics: Determinants and consequences” R&R

Manuscript number: PONE-D-24-10529

In the (revised version of this)manuscript, the authors assess some factors influencing affiliative needs and compliance with public health guidelines during the COVID-19 pandemic in France.

The manuscript presents an interesting case study of the circumstances and cultural changes that occurred during the COVID-19 pandemic. I found it very interesting, and the revisions that the authors have made to it have greatly improved its quality. However, I think a few things need to be further addressed. I, therefore, recommend a minor revision of this manuscript.

Below I provide details about my concerns, enumerated for ease of communication during the review process.

1. I appreciated the time and revisions that the authors have put into the introduction thus far. However, I think a little more space in the introduction is needed to describe horizontal social comparison. The paragraph describing this work (lines 59-65) is a little underwhelming. The only suggestion here that I have would be to unpack a little more information about how horizontal comparison is thought to reduce anxiety and uncertainty as well as the work on emergency affiliations (I think I can make an educated guess about how the variables are related, but it would be helpful to have this made more explicit for those who are not aware of the horizontal aspect of social comparison).

2. Lines 82-83 - it would be helpful to make the definition of social comparison more explicit that social comparison is about relying on what others think, feel, and how they behave so that individuals have more information about what appropriate behavior would be.

3. Lines 87 - 90. The results of these findings feel underexplained. I struggled a little to see the connection to the rest of the paragraph (again, I could make an educated guess about what you were insinuating, but it would be helpful to make this explicit).

4. I think the explanation for using the term gender in the introduction is still underdeveloped (Lines 91 - 107). The explanation in the response to reviewers and the discussion is much better. I would suggest rewriting this to me more like these sections (because you make it clearer why gender is the more appropriate term, instead of saying, “we used the terms because other people were not aware of the difference).

5. I find the method section to be very well done and clear. I appreciate the effort that the authors have put into revising this section.

6. Line 278. You refer to the bolded pink lines in Figure 1, but figure 1 does not have bolded pink lines. Do you mean figure 2?

7. With regard to figures, I greatly appreciate the revisions to these. They are very clear and easy to understand. The color coding and new line labels really make it easier to understand the results.

8. Line 335, note subheading. I’m not sure that the subheading is necessary, since the paragraph that follows it feels clearly like a method/ result paragraph.

9. Line 403. There seems to be a labeling error, so that there are two table 1’s. This table should be table 2 (which also makes the following tables incorrectly ordered as well).

10. I appreciate the effort that the authors have put into rewriting their results. The summary paragraphs greatly increase the interpretability of the results to a more novice audience.

11. I appreciate the revisions to the limitation section, these make the paper feel more whole and give many important considerations for future research.

12. Looking at the supplemental materials, I find these much easier to interpret and very helpful.

As per my policy, I sign all of my reviews. The author should feel free to contact me if they have any questions or points of clarification regarding our review.

Jessica D. Ayers

jessicaayers@boisestate.edu

6. PLOS authors have the option to publish the peer review history of their article (what does this mean?). If published, this will include your full peer review and any attached files.

Reviewer #1: No

Reviewer #2: **Yes: **Jessica D Ayers

---

## [Author Response · Author response to Decision Letter 0]

11 Jun 2024

This has been done.

2. Thank you for stating the following financial disclosure: "This work was supported by the French ANR (ANR-Flash COVID-19)." 

This has been done.

3. Please expand the acronym “ANR” (as indicated in your financial disclosure) so that it states the name of your funders in full.

This has been done.

This has been done.

Reviewer #1: I have been asked to treat this submission as a resubmission of a previous submission (manuscript ID PONE-D-22-34262). The comments that I raised in that previous version have been satisfactorily addressed.

We thank Reviewer #1 for their comments, which much improved our manuscript.

Reviewer #2: In the (revised version of this) manuscript, the authors assess some factors influencing affiliative needs and compliance with public health guidelines during the COVID-19 pandemic in France. The manuscript presents an interesting case study of the circumstances and cultural changes that occurred during the COVID-19 pandemic. I found it very interesting, and the revisions that the authors have made to it have greatly improved its quality. However, I think a few things need to be further addressed. I, therefore, recommend a minor revision of this manuscript. Below I provide details about my concerns, enumerated for ease of communication during the review process. 

We thank Reviewer #2 for a diligent reading and their new comments. We replied to them below. We also made the changes accordingly in the revised version of the manuscript.

1. I appreciated the time and revisions that the authors have put into the introduction thus far. However, I think a little more space in the introduction is needed to describe horizontal social comparison. The paragraph describing this work (lines 59-65) is a little underwhelming. The only suggestion here that I have would be to unpack a little more information about how horizontal comparison is thought to reduce anxiety and uncertainty as well as the work on emergency affiliations (I think I can make an educated guess about how the variables are related, but it would be helpful to have this made more explicit for those who are not aware of the horizontal aspect of social comparison).

We expanded this part, see lines 53-75.

2. Lines 82-83 - it would be helpful to make the definition of social comparison more explicit that social comparison is about relying on what others think, feel, and how they behave so that individuals have more information about what appropriate behavior would be.

We found your wording excellent and took the liberty to paraphrase your suggestion (see lines 54-56). Thank you!

3. Lines 87 - 90. The results of these findings feel underexplained. I struggled a little to see the connection to the rest of the paragraph (again, I could make an educated guess about what you were insinuating, but it would be helpful to make this explicit).

We rephrased this paragraph to improve its coherence (see lines 97-105).

4. I think the explanation for using the term gender in the introduction is still underdeveloped (Lines 91 - 107). The explanation in the response to reviewers and the discussion is much better. I would suggest rewriting this to me more like these sections (because you make it clearer why gender is the more appropriate term, instead of saying, “we used the terms because other people were not aware of the difference).

Thanks for this suggestion. We reckoned our sample would likely have not made a difference, which we believe would have been critical. This is explained lines 118-127.

5. I find the method section to be very well done and clear. I appreciate the effort that the authors have put into revising this section.

Thank you!

6. Line 278. You refer to the bolded pink lines in Figure 1, but figure 1 does not have bolded pink lines. Do you mean figure 2?

You are right – this has been corrected. Thank you!

7. With regard to figures, I greatly appreciate the revisions to these. They are very clear and easy to understand. The color coding and new line labels really make it easier to understand the results.

Thank you!

8. Line 335, note subheading. I’m not sure that the subheading is necessary, since the paragraph that follows it feels clearly like a method/ result paragraph.

We removed the subheading – thanks for the suggestion!

9. Line 403. There seems to be a labeling error, so that there are two table 1’s. This table should be table 2 (which also makes the following tables incorrectly ordered as well).

This has been corrected – thanks.

10. I appreciate the effort that the authors have put into rewriting their results. The summary paragraphs greatly increase the interpretability of the results to a more novice audience.

Thanks again for helping us!

11. I appreciate the revisions to the limitation section, these make the paper feel more whole and give many important considerations for future research.

Thank you!

12. Looking at the supplemental materials, I find these much easier to interpret and very helpful.

Thank you!

---

## [Editor Report · Decision Letter 1]

17 Jun 2024

Affiliation in times of pandemics: Determinants and consequences

PONE-D-24-10529R1

Dear Dr. Dezecache,

We’re pleased to inform you that your manuscript has been judged scientifically suitable for publication and will be formally accepted for publication once it meets all outstanding technical requirements.

Kind regards,

Pat Barclay

Academic Editor

PLOS ONE
---

## [Editor Report · Acceptance letter]

15 Aug 2024

PONE-D-24-10529R1 

PLOS ONE

Dear Dr. Dezecache, 

I'm pleased to inform you that your manuscript has been deemed suitable for publication in PLOS ONE. Congratulations! Your manuscript is now being handed over to our production team.

Kind regards, 

on behalf of

Dr. Pat Barclay 

Academic Editor

PLOS ONE